# Property (A) and Oscillation of Higher-Order Trinomial Differential Equations with Retarded and Advanced Arguments

## Blanka Baculikova

Department of Mathematics and Theoretical Informatics, Faculty of Electrical Engineering and Informatics, Technical University of Košice, Letná 9, 042 00 Košice, Slovakia; blanka.baculikova@tuke.sk

**Abstract:** In this paper, a new effective technique for the investigation of the higher-order trinomial differential equations $y^{(n)}(t) + p(t)y(\tau(t)) + q(t)y(\sigma(t)) = 0$ is established. We offer new criteria for so-called property (A) and oscillation of the considered equation. Examples are provided to illustrate the importance of our results.

**Keywords:** higher-order differential equations; retarded and advanced argument; oscillation; property (A); trinomial equation

**MSC:** 34K11; 34C10

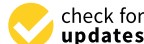



## 1. Introduction

Consider the functional differential equation of the form

$$y^{(n)}(t) + p(t)y(\tau(t)) + q(t)y(\sigma(t)) = 0, \tag{1}$$

where $n$ is odd or even number and the following conditions are assumed to hold:

$(H_1)$ $p(t), q(t) \in C([t_0, \infty))$, $p(t) > 0, q(t) > 0$,
$(H_2)$ $\tau(t) \in C^1([t_0, \infty))$, $\tau'(t) > 0$, $\tau(t) \le t$, $\lim_{t \to \infty} \tau(t) = \infty$,
$(H_3)$ $\sigma(t) \in C^1([t_0, \infty))$, $\sigma'(t) > 0$, $\sigma(t) \ge t$.

Usually, by a solution of Equation (1), we mean a function $y : [T_y, \infty) \to R$ which satisfies (1) for all sufficiently large $t$ and $\sup\{|y(t)| : t \ge T\} > 0$ for all $T \ge T_y$.

The oscillatory character of the solutions is understood in the standard way, that is, a proper solution is termed oscillatory or nonoscillatory according to whether it does or does not have infinitely many zeros.

In recent years, there has been increasing interest in studying the oscillation of solutions to different classes of differential equations, see e.g., [1–15]. This is due to the fact that they have numerous applications in natural sciences and engineering—see, for instance, the papers [13,14] for models from mathematical biology where oscillation and/or delay actions may be formulated by means of cross-diffusion terms.

By the well-known result of Kiguradze [5] (Lemma 1), one can easily classify the possible nonoscillatory solutions of (1). As a matter of fact, the set $\mathcal{N}$ of all nonoscillatory solutions of (1) has the following decomposition

$$\mathcal{N} = \mathcal{N}_0 \cup \mathcal{N}_2 \cup \cdots \cup \mathcal{N}_{n-1}, \quad n \text{ odd},$$
$$\mathcal{N} = \mathcal{N}_1 \cup \mathcal{N}_3 \cup \cdots \cup \mathcal{N}_{n-1}, \quad n \text{ even}.$$

where $y(t) \in \mathcal{N}_\ell$ means that there exists $t_0 \ge T_y$ such that

$$\begin{aligned} y(t)y^{(i)}(t) &> 0 \quad \text{on } [t_0, \infty) \text{ for } 0 \le i \le \ell, \\ (-1)^i y(t)y^{(i)}(t) &> 0 \quad \text{on } [t_0, \infty) \text{ for } \ell \le i \le n. \end{aligned} \tag{2}$$

Such a $y(t)$ is said to be a solution of degree $\ell$.

Following the classical results of Kiguradze [5], we say that Equation (1) enjoys property $(A)$ if

$$\mathcal{N} = \mathcal{N}_0, \quad n \text{ odd},$$
$$\mathcal{N} = \varnothing, \quad n \text{ even}.$$

This definition formulates the fact that (1) with $\tau(t) = \sigma(t) \equiv t$ and $n$ odd always possesses a solution of degrees 0 that is $\mathcal{N}_0 \neq \varnothing$ in this case.

The initial effort of mathematicians was oriented towards establishing criteria for property (A) of (1), which means to empty all classes $\mathcal{N}_\ell$ for all $\ell \neq 0$.

We recall the excellent criteria of Koplatadze et al. [9] that have been formulated for binomial differential equations.

**Theorem 1.** (Theorem 2.1 in [9]). *If $q(t) \equiv 0$ and*

$$\limsup_{t \to \infty} \left\{ \tau(t) \int_t^\infty [\tau(s)]^{n-2} p(s) \, ds + \int_{\tau(t)}^t [\tau(s)]^{n-1} p(s) \, ds \right.$$
$$\left. + \frac{1}{\tau(t)} \int_0^{\tau(t)} s[\tau(s))]^{n-1} p(s) \, ds \right\} > (n-1)!, \tag{3}$$

*then (1) has property (A).*

**Theorem 2.** (Theorem 2.2 in [9]). *If $p(t) \equiv 0$, $n$ is even and*

$$\limsup_{t \to \infty} \left\{ \sigma(t) \int_{\sigma(t)}^\infty s^{n-2} p(s) \, ds + \int_t^{\sigma(t)} s^{n-1} p(s) \, ds \right.$$
$$\left. + \frac{1}{\sigma(t)} \int_0^t s^{n-1} \sigma(t) p(s) \, ds \right\} > (n-1)!, \tag{4}$$

*then (1) has property (A).*

**Theorem 3.** (Theorem 2.3 in [9]). *If $p(t) \equiv 0$, $n$ is odd and*

$$\limsup_{t \to \infty} \left\{ \sigma(t) \int_{\sigma(t)}^\infty \sigma(s) s^{n-3} p(s) \, ds + \int_t^{\sigma(t)} s^{n-2} \sigma(s) p(s) \, ds \right.$$
$$\left. + \frac{1}{\sigma(t)} \int_0^t s^{n-2} [\sigma(t)]^2 p(s) \, ds \right\} > 2(n-2)!, \tag{5}$$

*and*

$$\limsup_{t \to \infty} \left\{ \sigma(t) \int_{\sigma(t)}^\infty [\sigma(s)]^{n-2} p(s) \, ds + \int_t^{\sigma(t)} s[\sigma(s)]^{n-2} p(s) \, ds \right.$$
$$\left. + \frac{1}{\sigma(t)} \int_0^t s[\sigma(t)]^{n-1} p(s) \, ds \right\} > (n-1)!, \tag{6}$$

*then (1) has property (A).*

The first aim of this paper is to extend the above-mentioned criteria known for binomial differential equations to more general trinomial equations. The second aim of this paper is to establish criteria for the class $\mathcal{N}_0 = \varnothing$, which leads to the oscillation of (1) also for $n$ odd, which are new phenomena for (1).

## 2. Results

As an auxiliary statements, we recall the following result from [9].

**Lemma 1.** *Let $y(t) \in \mathcal{N}_\ell$ for some $\ell \in \{1, \cdots, n-1\}$ and*

$$\int^\infty t^{n-\ell} |y^{(n)}(t)| \, dt = \infty. \tag{7}$$

*Then there exists $t_* \geq t_0$ such that for $t \geq t_*$*

$$\frac{y(t)}{t^\ell} \downarrow, \qquad \frac{y(t)}{t^{\ell-1}} \uparrow \tag{8}$$

*and*

$$y(t) \geq \frac{t^\ell}{\ell!(n-\ell)!} \int_t^\infty \left( s^{n-\ell-1} (p(s)y(\tau(s)) + q(s)y(\sigma(s))) \right) ds$$
$$+ \frac{t^{\ell-1}}{\ell!(n-\ell)!} \int_{t_*}^t \left( s^{n-\ell} (p(s)y(\tau(s)) + q(s)y(\sigma(s))) \right) ds. \tag{9}$$

Moreover, if $y(t) \in \mathcal{N}_\ell$, then $y(t) \geq c t^{\ell-1}$ for some $c > 0$. Since

$$t^{n-\ell} |y^{(n)}(t)| = t^{n-\ell} (p(t)y(\tau(t)) + q(t)y(\sigma(t)))$$
$$\geq c t^{n-\ell} \left( p(t)\tau^{\ell-1}(t) + q(t)\sigma^{\ell-1}(t) \right) \geq c \left( p(t)\tau^{n-1}(t) + q(t)t^{n-1} \right),$$

we can replace nonstandard condition (7) by the following easily verifiable one.

$$\int^\infty \left( \tau^{n-1}(s)p(s) + s^{n-1}q(s) \right) ds = \infty. \tag{10}$$

which follows from the fact that if $y(t) \in \mathcal{N}_\ell$, then $y(t) \geq c t^{\ell-1}$, $c > 0$.

Now, we are prepared to formulate the first criterion for property (A) of (1).

**Theorem 4.** *Assume that for n even*

$$\limsup_{t \to \infty} \left\{ \int_{\tau(t)}^t \tau^{n-1}(s)p(s)\,ds + \tau(t) \int_t^\infty \tau^{n-2}(s)p(s) + s^{n-2}q(s)\,ds \right.$$
$$\left. + \tau(t) \int_{\tau(t)}^t s^{n-2}q(s)\,ds + \frac{1}{\tau(t)} \int_{t_*}^{\tau(t)} s\tau^{n-1}(s)p(s) + s^n q(s)\,ds \right\} > (n-1)!, \tag{11}$$

*and for n odd*

$$\limsup_{t \to \infty} \left\{ \int_{\tau(t)}^t \tau^{n-1}(s)p(s)\,ds + \tau(t) \int_t^\infty \tau^{n-2}(s)p(s) + s^{n-3}\sigma(s)q(s)\,ds \right.$$
$$+ \tau(t) \int_{\tau(t)}^t s^{n-3}\sigma(s)q(s)\,ds + \frac{1}{\tau(t)} \int_{t_*}^{\tau(t)} s\tau^{n-1}(s)p(s) + s^{n-1}\sigma(s)q(s)\,ds \left. \right\} > (n-1)!, \tag{12}$$

*then (1) has property (A).*

**Proof.** Suppose, to contrary, that $y(t)$ is an eventually positive solution of (1) such that $y(t) \in \mathcal{N}_\ell$ for some $\ell \in \{1, \cdots, n-1\}$ such that $n + \ell$ is odd. To be able to use results of Lemma 1, we shall show that (11) and (12) implies (10). Really, if we admit that

$$\int^\infty \left( \tau^{n-1}(s)p(s) + s^{n-1}q(s) \right) ds < \infty,$$

then for given $\varepsilon > 0$, there exists a $t_1$ such that

$$\int_{t_1}^{\infty} \left( \tau^{n-1}(s)p(s) + s^{n-1}q(s) \right) ds < \varepsilon,$$

If we consider $\tau(t) \geq t_1$, then

$$\int_{\tau(t)}^{t} \tau^{n-1}(s)p(s)\, ds + \tau(t) \int_{\tau(t)}^{t} s^{n-3}\sigma(s)q(s)\, ds$$

$$\leq \int_{\tau(t)}^{t} \left( \tau^{n-1}(s)p(s) + s^{n-1}q(s) \right) ds < \varepsilon. \tag{13}$$

On the other hand, for $t \geq t_1$

$$\tau(t) \int_{t}^{\infty} \tau^{n-2}(s)p(s) + s^{n-2}\sigma(s)q(s)\, ds \leq \int_{t}^{\infty} \left( \tau^{n-1}(s)p(s) + s^{n-1}q(s) \right) ds < \varepsilon, \tag{14}$$

and finally for $t_* \geq t_1$

$$\frac{1}{\tau(t)} \int_{t_*}^{\tau(t)} s\tau^{n-1}(s)p(s) + s^n q(s)\, ds \leq \int_{t_*}^{\tau(t)} \left( \tau^{n-1}(s)p(s) + s^{n-1}q(s) \right) ds < \varepsilon. \tag{15}$$

Combining (13)–(15) one gets

$$\lim_{t \to \infty} \left\{ \int_{\tau(t)}^{t} \tau^{n-1}(s)p(s)\, ds + \tau(t) \int_{t}^{\infty} \tau^{n-2}(s)p(s) + s^{n-2}q(s)\, ds \right.$$

$$\left. + \tau(t) \int_{\tau(t)}^{t} s^{n-2}q(s)\, ds + \frac{1}{\tau(t)} \int_{t_*}^{\tau(t)} s\tau^{n-1}(s)p(s) + s^n q(s)\, ds \right\} = 0.$$

This contradicts (11) and we conclude that (10) holds true. Therefore, taking into account (9), we get

$$y(\tau(t)) \geq \frac{\tau^{\ell}(t)}{(n-\ell)!\ell!} \int_{\tau(t)}^{t} s^{n-\ell-1}p(s)\frac{y(\tau(s))}{\tau^{\ell}(s)}\tau^{\ell}(s) + s^{n-\ell-1}q(s)\frac{y(\sigma(s))}{\sigma^{\ell-1}(s)}\sigma^{\ell-1}(s)\, ds$$

$$+ \frac{\tau^{\ell}(t)}{(n-\ell)!\ell!} \int_{t}^{\infty} s^{n-\ell-1}p(s)\frac{y(\tau(s))}{\tau^{\ell-1}(s)}\tau^{\ell-1}(s) + s^{n-\ell-1}q(s)\frac{y(\sigma(s))}{\sigma^{\ell-1}(s)}\sigma^{\ell-1}(s)\, ds$$

$$+ \frac{\tau^{\ell-1}(t)}{(n-\ell)!\ell!} \int_{t_*}^{\tau(t)} s^{n-\ell}p(s)\frac{y(\tau(s))}{\tau^{\ell}(s)}\tau^{\ell}(s) + s^{n-\ell}q(s)\frac{y(\sigma(s))}{\sigma^{\ell-1}(s)}\sigma^{\ell-1}(s)\, ds.$$

Employing (8), that is

$$\frac{y(\tau(t))}{\tau^{\ell}(t)} \downarrow, \qquad \frac{y(\tau(t))}{\tau^{\ell-1}(t)} \uparrow, \qquad \frac{y(\sigma(t))}{\sigma^{\ell-1}(t)} \geq \frac{y(t)}{t^{\ell-1}} \geq \frac{y(\tau(t))}{\tau^{\ell-1}(t)},$$

we are led to

$$(n-\ell)!\ell!y(\tau(t)) \geq y(\tau(t)) \int_{\tau(t)}^{t} s^{n-\ell-1}\tau^{\ell}(s)p(s)\, ds + \tau^{\ell}(t) \int_{\tau(t)}^{t} s^{n-\ell-1}\sigma^{\ell-1}(s)\frac{y(s)}{s^{\ell-1}}q(s)\, ds$$

$$+ \tau(t)y(\tau(t)) \int_{t}^{\infty} s^{n-\ell-1}\tau^{\ell-1}(s)p(s)\, ds + \tau^{\ell}(t) \int_{t}^{\infty} s^{n-\ell-1}\sigma^{\ell-1}(s)\frac{y(\tau(s))}{\tau^{\ell-1}(s)}q(s)\, ds$$

$$+ \frac{y(\tau(t))}{\tau(t)} \int_{t_*}^{\tau(t)} s^{n-\ell}\tau^{\ell}(s)p(s)\, ds + \tau^{\ell-1}(t) \int_{t_*}^{\tau(t)} s^{n-\ell+1}\sigma^{\ell-1}(s)\frac{y(s)}{s^{\ell}}q(s)\, ds.$$

Therefore, we get

$$
\begin{aligned}
(n-\ell)!\ell! \geq & \int_{\tau(t)}^{t} s^{n-\ell-1}\tau^{\ell}(s)p(s)\,\mathrm{d}s + \tau(t)\int_{\tau(t)}^{t} s^{n-\ell-1}\sigma^{\ell-1}(s)q(s)\,\mathrm{d}s \\
& + \tau(t)\int_{t}^{\infty} s^{n-\ell-1}\tau^{\ell-1}(s)p(s)\,\mathrm{d}s + \tau(t)\int_{t}^{\infty} s^{n-\ell-1}\sigma^{\ell-1}(s)q(s)\,\mathrm{d}s \\
& + \frac{1}{\tau(t)}\int_{t_*}^{\tau(t)} s^{n-\ell}\tau^{\ell}(s)p(s)\,\mathrm{d}s + \frac{1}{\tau(t)}\int_{t_*}^{\tau(t)} s^{n-\ell+1}\sigma^{\ell-1}(s)q(s)\,\mathrm{d}s.
\end{aligned}
\tag{16}
$$

The following considerations are intended to eliminate parameter $\ell$. We use the fact that $\tau(t) \leq t \leq \sigma(t)$. We shall distinguish the parity of $n$. For $n$ even, we have $\ell \in \{1, 3, \ldots, n-1\}$, and it follows from (16) that

$$
(n-1)! \geq \int_{\tau(t)}^{t} \tau^{n-1}(s)p(s)\,\mathrm{d}s + \tau(t)\int_{\tau(t)}^{t} s^{n-2}q(s)\,\mathrm{d}s
$$
$$
+ \tau(t)\int_{t}^{\infty} \tau^{n-2}(s)p(s) + s^{n-2}q(s)\,\mathrm{d}s + \frac{1}{\tau(t)}\int_{t_*}^{\tau(t)} s\tau^{n-1}(s)p(s) + s^{n}q(s)\,\mathrm{d}s.
$$

The last inequality contradicts (11).

Now, we consider $n$ even. Then, $\ell \in \{2, 4, \ldots, n-1\}$ and (16) yields

$$
(n-1)! \geq \int_{\tau(t)}^{t} \tau^{n-1}(s)p(s)\,\mathrm{d}s + \tau(t)\int_{\tau(t)}^{t} s^{n-3}\sigma(s)q(s)\,\mathrm{d}s
$$
$$
+ \tau(t)\int_{t}^{\infty} \tau^{n-2}(s)p(s) + s^{n-3}\sigma(s)q(s)\,\mathrm{d}s + \frac{1}{\tau(t)}\int_{t_*}^{\tau(t)} s\tau^{n-1}(s)p(s) + s^{n-1}\sigma(s)q(s)\,\mathrm{d}s.
$$

The last inequality contradicts (12) and the proof is complete. $\square$

The main idea of the proof of Theorem 4 is to nominate delay argument $\tau(t)$ into (9). Now, we are about to substitute advanced argument $\sigma(t)$ into (9) to obtain another (independent) criterion for property (A).

**Theorem 5.** *Assume that for n even*

$$
\begin{aligned}
\limsup_{t\to\infty}\Bigg\{ & \frac{1}{\sigma(t)}\int_{t}^{\sigma(t)} s\tau^{n-1}(s)p(s)\,\mathrm{d}s + \sigma(t)\int_{\sigma(t)}^{\infty} \frac{\tau^{n-1}(s)}{s}p(s) + s^{n-2}q(s)\,\mathrm{d}s \\
& + \int_{t}^{\sigma(t)} s^{n-1}q(s)\,\mathrm{d}s + \frac{1}{\sigma(t)}\int_{t_*}^{t} s\tau^{n-1}(s)p(s) + \sigma(s)s^{n-1}q(s)\,\mathrm{d}s \Bigg\} > (n-1)!,
\end{aligned}
\tag{17}
$$

*and for n odd*

$$
\begin{aligned}
\limsup_{t\to\infty}\Bigg\{ & \frac{1}{\sigma(t)}\int_{t}^{\sigma(t)} s\tau^{n-1}(s)p(s)\,\mathrm{d}s + \sigma(t)\int_{\sigma(t)}^{\infty} \frac{\tau^{n-1}(s)}{s}p(s) + \sigma(s)s^{n-3}q(s)\,\mathrm{d}s \\
& + \int_{t}^{\sigma(t)} \sigma(s)s^{n-2}q(s)\,\mathrm{d}s + \frac{1}{\sigma(t)}\int_{t_*}^{t} s\tau^{n-1}(s)p(s) + \sigma^{2}(s)s^{n-2}q(s)\,\mathrm{d}s \Bigg\} > (n-1)!,
\end{aligned}
\tag{18}
$$

*then* (1) *has property (A).*

**Proof.** Suppose, to contrary, that (1) does not enjoy property (A). This means that (1) possesses an eventually positive solution $y(t) \in \mathcal{N}_\ell$ for some $\ell \in \{1, \cdots, n-1\}$ with $n + \ell$ odd. It follows from (9) that

$$y(\sigma(t)) \geq \frac{\sigma^\ell(t)}{(n-\ell)!\ell!} \int_{\sigma(t)}^\infty s^{n-\ell-1} p(s) \frac{y(\tau(s))}{\tau^\ell(s)} \tau^\ell(s) + s^{n-\ell-1} q(s) \frac{y(\sigma(s))}{\sigma^{\ell-1}(s)} \sigma^{\ell-1}(s) \, ds$$

$$+ \frac{\sigma^{\ell-1}(t)}{(n-\ell)!\ell!} \int_t^{\sigma(t)} s^{n-\ell} p(s) \frac{y(\tau(s))}{\tau^\ell(s)} \tau^\ell(s) + s^{n-\ell} q(s) \frac{y(\sigma(s))}{\sigma^{\ell-1}(s)} \sigma^{\ell-1}(s) \, ds$$

$$+ \frac{\sigma^{\ell-1}(t)}{(n-\ell)!\ell!} \int_{t_*}^t s^{n-\ell} p(s) \frac{y(\tau(s))}{\tau^\ell(s)} \tau^\ell(s) + s^{n-\ell} q(s) \frac{y(\sigma(s))}{\sigma^\ell(s)} \sigma^\ell(s) \, ds.$$

Condition (8) implies that

$$\frac{y(\sigma(t))}{\sigma^\ell(t)} \downarrow, \qquad \frac{y(\sigma(t))}{\sigma^{\ell-1}(t)} \uparrow, \qquad \frac{y(\tau(t))}{\tau^\ell(t)} \geq \frac{y(t)}{t^\ell} \geq \frac{y(\sigma(t))}{\sigma^\ell(t)},$$

Therefore,

$$(n-1)! y(\sigma(t)) \geq \sigma^\ell(t) \int_{\sigma(t)}^\infty s^{n-\ell-1} p(s) \frac{y(s)}{s^\ell(s)} \tau^\ell(s) \, ds + \sigma(t) y(\sigma(t)) \int_{\sigma(t)}^\infty s^{n-\ell-1} q(s) \sigma^{\ell-1}(s) \, ds$$

$$+ \sigma^{\ell-1}(t) \int_t^{\sigma(t)} s^{n-\ell} p(s) \frac{y(s)}{s^\ell(s)} \tau^\ell(s) \, ds + y(\sigma(t)) \int_t^{\sigma(t)} s^{n-\ell} q(s) \sigma^{\ell-1}(s) \, ds$$

$$+ \sigma^{\ell-1}(t) \int_{t_*}^t s^{n-\ell} p(s) \frac{y(s)}{s^\ell(s)} \tau^\ell(s) \, ds + \frac{y(\sigma(s))}{\sigma(s)} \int_{t_*}^t s^{n-\ell} q(s) \sigma^\ell(s) \, ds.$$

Employing (8), one gets

$$(n-1)! \geq \sigma(t) \int_{\sigma(t)}^\infty s^{n-\ell-2} \tau^\ell(s) p(s) + s^{n-\ell-1} \sigma^{\ell-1}(s) q(s) \, ds$$

$$+ \frac{1}{\sigma(t)} \int_t^{\sigma(t)} s^{n-\ell} \tau^\ell(s) p(s) \, ds + \int_t^{\sigma(t)} s^{n-\ell} \sigma^{\ell-1}(s) q(s) \, ds \qquad (19)$$

$$+ \frac{1}{\sigma(t)} \int_{t_*}^t s^{n-\ell} \tau^\ell(s) p(s) \, ds + s^{n-\ell} \sigma^\ell(s) q(s) \, ds.$$

For $n$ even we have $\ell \in \{1, 3, \ldots, n-1\}$ and (19) in view of $\tau(t) \leq t \leq \sigma(t)$ implies

$$(n-1)! \geq \sigma(t) \int_{\sigma(t)}^\infty \frac{\tau^{n-1}(s)}{s} p(s) + s^{n-2} q(s) \, ds + \frac{1}{\sigma(t)} \int_t^{\sigma(t)} s \tau^{n-1}(s) p(s) \, ds$$

$$+ \int_t^{\sigma(t)} s^{n-1} q(s) \, ds + \frac{1}{\sigma(t)} \int_{t_*}^t s \tau^{n-1}(s) p(s) \, ds + s^{n-1} \sigma(s) q(s) \, ds$$

which contradicts (17).

On the other hand, for $n$ even, we have $\ell \in \{2, 4, \ldots, n-1\}$, and now (19) yields

$$(n-1)! \geq \sigma(t) \int_{\sigma(t)}^\infty \frac{\tau^{n-1}(s)}{s} p(s) + \sigma(s) s^{n-3} q(s) \, ds + \frac{1}{\sigma(t)} \int_t^{\sigma(t)} s \tau^{n-1}(s) p(s) \, ds$$

$$+ \int_t^{\sigma(t)} \sigma(s) s^{n-2} q(s) \, ds + \frac{1}{\sigma(t)} \int_{t_*}^t s \tau^{n-1}(s) p(s) \, ds + s^{n-2} \sigma^2(s) q(s) \, ds.$$

This contradicts (18), and the proof is finished. $\square$

The following example is intended to show that the criteria for property (A) presented in Theorems 4 and 5 are independent.

**Example 1.** *Consider the equation*

$$y^{(n)}(t) + \frac{p_0}{t^n} y(\lambda t) + \frac{q_0}{t^n} y\left(\frac{1}{\lambda} t\right) = 0, \tag{20}$$

*where $p_0 > 0$, $q_0 > 0$, $\lambda \in (0,1)$.*

*Easy computations show that conditions (11) and (17) applied to (20) reduce to*

$$p_0 \lambda^{n-1} (2 - \ln \lambda) + 2q_0 > (n-1)! \tag{21}$$

*and*

$$2p_0 \lambda^{n-1} + q_0 (2 - \ln \lambda) > (n-1)!, \tag{22}$$

*respectively. It is easy that (21) and (22) are independent, since*

$$p_0 \lambda^{n-1} (2 - \ln \lambda) > 2p_0 \lambda^{n-1} \quad and \quad 2q_0 < q_0 (2 - \ln \lambda).$$

*On the other hand, conditions (12) and (18) applied to (20) take the form*

$$p_0 \lambda^{n-1} (2 - \ln \lambda) + \frac{2q_0}{\lambda} > (n-1)! \tag{23}$$

*and*

$$2p_0 \lambda^{n-1} + \frac{2q_0}{\lambda} (2 - \ln \lambda) > (n-1)!, \tag{24}$$

*respectively, and again, (23) and (24) are independent. Therefore, by Theorems 4 and 5 if*
*(i)   for n even (21) or (22) holds,*
*(ii)  for n odd (23) or (24) is satisfied,*
*then, (20) has property (A). For the particular case when $n = 3$, $\lambda = 0.5$ criterion (24) is satisfied, e.g., for*

$$p_0 = q_0 > 0.1774.$$

Now, we turn our attention to the class $\mathcal{N}_0$, where of course $n$ is odd. It is easy to see that $(H_2)$ guaranties the existence of the inverse function $\tau^{-1}(t)$, and therefore, the auxiliary function $\xi(t) \in C^1([t_0, \infty))$

$$\xi(\xi(t)) = \tau^{-1}(t). \tag{25}$$

is well defined. The following lemma is elementary but very useful in our next considerations.

**Lemma 2.** *Assume that $\xi(t)$ satisfies (25). Then*

$$\xi^{-1}(\xi^{-1}(t)) = \tau(t), \quad \xi(t) = \tau^{-1}(\xi^{-1}(t)), \quad \xi^{-1}(t) = \tau(\xi(t)). \tag{26}$$

Now, our aim is to establish a criterion for $\mathcal{N}_0 = \varnothing$ for (1) for $n$ odd. To simplify our notation, we employ the following functions:

$$
\begin{aligned}
P_1(t) &= \int_t^{\xi(t)} \frac{(s-t)^{n-1}}{(n-1)!} p(s) \, \mathrm{d}s, \\
P_2(t) &= \int_{\xi(t)}^{\tau^{-1}(t)} \frac{(s-t)^{n-1}}{(n-1)!} p(s) \, \mathrm{d}s, \\
P_3(t) &= \int_{\tau^{-1}(t)}^{\tau^{-1}(\xi(t))} \frac{(s-t)^{n-1}}{(n-1)!} p(s) \, \mathrm{d}s.
\end{aligned}
\tag{27}
$$

and

$$Q_1(t) = \int_t^{\xi(t)} \frac{(s-t)^{n-1}}{(n-1)!} q(s)\, ds,$$

$$Q_2(t) = \int_{\xi(t)}^{\tau^{-1}(t)} \frac{(s-t)^{n-1}}{(n-1)!} q(s)\, ds.$$

(28)

We assume that there exist positive constants $P_i$, $i = 1, 2, 3$ and $Q_i$, $i = 1, 2$ such that

$$P_i(t) \geq P_i, \quad i = 1, 2, 3 \quad \text{and} \quad Q_i(t) \geq Q_i, \quad i = 1, 2.$$

Moreover, we set

$$P_i^* = \frac{P_i}{1 - P_2}, \quad i = 1, 3, \quad Q_i^* = \frac{Q_i}{1 - P_2}, \quad i = 1, 2,$$

and

$$M = \frac{(P_1^*)^2}{1 - P_1^* P_3^*}, \quad N = \frac{P_1^*}{1 - P_1^* P_3^* - P_1^* Q_1^* M}.$$

**Theorem 6.** *Assume that there exists a function $\xi(t)$ satisfying (25) and $\tau^{-1}(t) \geq \sigma(t)$. If*

$$P_3^* N + MN(Q_1^* + P_1^* Q_2^*) + P_1^* \left[ P_1^* + P_3^* + P_1^* Q_1^* N + Q_2^* MN \right] > 1,$$

(29)

*then $\mathcal{N}_0 = \varnothing$ for (1) for $n$ odd.*

**Proof.** Suppose, to contrary, that $y(t)$ is an eventually positive solution of (1) for $n$ odd such that $y(t) \in \mathcal{N}_0$. Integrating (1) twice from $t$ to $\infty$ and changing the order of integration, we are led to

$$-y^{(n-2)}(t) \geq \int_t^\infty (s-t)p(s)y(\tau(s)) + (s-t)q(s)y(\sigma(s))\, ds.$$

(30)

By repeated integration above inequality from $t$ to $\infty$ and changing the order of integration, we get

$$\begin{aligned} y(t) &\geq \int_t^\infty \frac{(s-t)^{n-1}}{(n-1)!} p(s)y(\tau(s)) + \frac{(s-t)^{n-1}}{(n-1)!} q(s)y(\sigma(s))\, ds \\ &\geq \int_t^\infty \frac{(s-t)^{n-1}}{(n-1)!} p(s)y(\tau(s)) + \frac{(s-t)^{n-1}}{(n-1)!} q(s)y(\tau^{-1}(s))\, ds. \end{aligned}$$

(31)

Consequently,

$$\begin{aligned} y(t) &\geq \int_t^{\xi(t)} \frac{(s-t)^{n-1}}{(n-1)!} p(s)y(\tau(s))\, ds + \int_{\xi(t)}^{\tau^{-1}(t)} \frac{(s-t)^{n-1}}{(n-1)!} p(s)y(\tau(s))\, ds \\ &+ \int_{\tau^{-1}(t)}^{\tau^{-1}(\xi(t))} \frac{(s-t)^{n-1}}{(n-1)!} p(s)y(\tau(s))\, ds + \int_t^{\xi(t)} \frac{(s-t)^{n-1}}{(n-1)!} q(s)y\left(\tau^{-1}(s)\right)\, ds \\ &+ \int_{\xi(t)}^{\tau^{-1}(t)} \frac{(s-t)^{n-1}}{(n-1)!} q(s)y\left(\tau^{-1}(s)\right)\, ds. \end{aligned}$$

(32)

Using the fact that $y(t)$ is a decreasing function, we are led to

$$y(t) \geq P_1 y(\xi^{-1}(t)) + P_2 y(t) + P_3 y(\xi(t)) + Q_1 y\left(\tau^{-1}(\xi(t))\right) + Q_2 y\left(\tau^{-1}\left(\tau^{-1}(t)\right)\right)$$

which is equivalent to

$$y(t) \geq P_1^* y(\xi^{-1}(t)) + P_3^* y(\xi(t)) + Q_1^* y\left(\tau^{-1}(\xi(t))\right) + Q_2^* y\left(\tau^{-1}\left(\tau^{-1}(t)\right)\right). \quad (33)$$

We are about to evaluate $y(t)$ in terms of $P_i^*$ and $Q_i^*$. For all that, we step by step set

$$t = \xi^{-1}(t), \quad t = \xi(t), \quad t = \tau^{-1}(\xi(t)), \quad t = \tau^{-1}\left(\tau^{-1}(t)\right)$$

into (33). Before executing all that, to simplify our notation, we set

$$\begin{aligned} Y =& y(t), \quad A = y(\xi^{-1}(t)), \quad B = y(\xi(t)), \quad C = y\left(\tau^{-1}(\xi(t))\right), \\ D =& y\left(\tau^{-1}\left(\tau^{-1}(t)\right)\right), \quad E = y(\tau^{-1}(t)). \end{aligned}$$

Finally, we are led to the following linear algebraic inequalities

$$\begin{aligned} Y &\geq P_1^* A + P_3^* B + Q_1^* C + Q_2^* D, \\ A &\geq (P_1^* + P_3^*)Y + Q_1^* E + Q_2^* C, \\ B &\geq P_1^* Y + P_3^* E + Q_1^* D, \\ C &\geq P_1^* E + P_3^* D, \\ D &\geq P_1^* C, \\ E &\geq P_1^* B. \end{aligned}$$

Elimination of variables $D$ and $E$ leads to

$$\begin{aligned} Y &\geq P_1^* A + P_3^* B + (Q_1^* + P_1^* Q_2^*)C, \\ A &\geq (P_1^* + P_3^*)Y + P_1^* Q_1^* B + Q_2^* C, \\ B &\geq P_1^* Y + P_1^* P_3^* B + P_1^* Q_1^* C, \\ C &\geq (P_1^*)^2 B + P_1^* P_3^* C. \end{aligned}$$

Thus,

$$C \geq \frac{(P_1^*)^2}{1 - P_1^* P_3^*} B = MB$$

which implies

$$B \geq \frac{P_1^*}{1 - P_1^* P_3^* - P_1^* Q_1^* M} Y = NY.$$

Consequently, our system reduces onto the couple of inequalities

$$\begin{aligned} Y &\geq P_1^* A + P_3^* NY + MN(Q_1^* + P_1^* Q_2^*)Y, \\ A &\geq (P_1^* + P_3^*)Y + P_1^* Q_1^* NY + Q_2^* MNY \end{aligned}$$

which means that

$$y(t)\left\{1 - P_3^* N - MN(Q_1^* + P_1^* Q_2^*) - P_1^*\left[P_1^* + P_3^* + P_1^* Q_1^* N + Q_2^* MN\right]\right\} > 0.$$

This contradicts (29), and we conclude that $\mathcal{N}_0 = \varnothing$. □

The following criteria follows immediately from the proof of Theorem 6.

**Corollary 1.** *Assume that there exists a function $\xi(t)$ satisfying (26) and $\tau^{-1}(t) \geq \sigma(t)$. If*

$$P_2 > 1, \quad or \quad P_1 P_3 + P_1 Q_1 M > 1,$$

*then $\mathcal{N}_0 = \varnothing$ for (1).*

**Example 2.** *Consider once more the equation*

$$y^{(n)}(t) + \frac{p_0}{t^n} y(\lambda t) + \frac{q_0}{t^n} y\left(\frac{1}{\lambda} t\right) = 0, \tag{34}$$

*where $p_0 > 0$, $q_0 > 0$, $\lambda \in (0, 1)$.*

*We have already presented criteria for property (A) of (20). If for n odd, we join on condition (29), we obtain oscillation of (20). It remains to evaluate constants $P_i$ and $Q_j$ for (20). We set $\xi(t) = t/\sqrt{\lambda}$. Now,*

$$P_1(t) = \frac{p_0}{(n-1)!} \int_t^{t/\sqrt{\lambda}} \frac{(s-t)^{n-1}}{s^n} \, ds = \left| x = 1 - \frac{t}{s} \right| = \frac{p_0}{(n-1)!} \int_0^{1-\sqrt{\lambda}} \frac{x^{n-1}}{1-x} \, dx$$

$$= \frac{p_0}{(n-1)!} \left( -\ln\sqrt{\lambda} - \sum_{i=1}^{n-1} \frac{\left(1-\sqrt{\lambda}\right)^i}{i} \right) = P_1.$$

*Similarly, we evaluate*

$$P_2 = \frac{p_0}{(n-1)!} \left( -\ln\sqrt{\lambda} - \sum_{i=1}^{n-1} \frac{(1-\lambda)^i - \left(1-\sqrt{\lambda}\right)^i}{i} \right),$$

$$P_3 = \frac{p_0}{(n-1)!} \left( -\ln\sqrt{\lambda} - \sum_{i=1}^{n-1} \frac{\left(1-\lambda\sqrt{\lambda}\right)^i - (1-\lambda)^i}{i} \right),$$

$$Q_1 = \frac{q_0}{(n-1)!} \left( -\ln\sqrt{\lambda} - \sum_{i=1}^{n-1} \frac{\left(1-\sqrt{\lambda}\right)^i}{i} \right),$$

$$Q_2 = \frac{q_0}{(n-1)!} \left( -\ln\sqrt{\lambda} - \sum_{i=1}^{n-1} \frac{(1-\lambda)^i - \left(1-\sqrt{\lambda}\right)^i}{i} \right).$$

*Employing mathematical software (Matlab R2019), we see that for the particular case when $n = 3$ and $\lambda = 0.5$, criterion (29) is satisfied, e.g., for*

$$p_0 = q_0 > 17.3$$

*which guarantees oscillation of (20) for this case. By the way, the well-known criterion of Koplatadze and Chanturija [10] (see also [11]) requires $p_0 > 29.3482$ to ensure oscillation of (20) for $n = 3$ and $\lambda = 0.5$. The progress is outstanding.*

## 3. Discussion

In this paper, we have established a new technique for the investigation of trinomial differential equations with retarded and advanced argument. First, we extended some results from binomial to trinomial differential equation. For the second, we introduced a new technique for classes $\mathcal{N}_0$ and $\mathcal{N}_n$ to be empty, which leads to the oscillation of (1). This fact has not be considered in [9]. The open problem remains how to extend our criteria for more general differential equations (nonlinear, neutral).

**Funding:** This research received no external funding.

**Institutional Review Board Statement:** Not applicable for studies not involving humans or animals.

**Informed Consent Statement:** Not applicable for studies not involving humans.

**Data Availability Statement:** The data presented in this study are available on request from the corresponding author.

**Conflicts of Interest:** The author declare no conflicts of interest.

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
