# Peer review of "Property (A) and Oscillation of Higher-Order Trinomial Differential Equations with Retarded and Advanced Arguments"

_mathematics, doi:10.3390/math12060910_

Round 1
Reviewer 1 Report
Comments and Suggestions for Authors
See the report, especially the question concerning Lemma A.

Reviewer 2 Report
Comments and Suggestions for Authors
Please see attachment.

Comments on the Quality of English LanguageModerate editing of English language required.
Reviewer 3 Report
Comments and Suggestions for Authors
Report
February 21st, 2024
Paper: Property (A) and oscillation of higher-order trinomial differential equations with retarded and advanced arguments
Blanka Baculikova
Mathematics-2840468 (paper number)
Mathematics
Summary of the paper from the reviewer
The topic of the paper is interesting. Here is provided a new effective technique for investigation of the higher order trinomial differential equations
y(n)(t)+p(t)y(T(t))+q(t)y(s(t))=0
Here is also, provided a criteria for oscillation of considered equation as well as examples which illustrate the importance of results.
In the introduction is not clearly enough presented achievement of this research. However, the introduction needs to make it more clear what the main objective of the paper is, and what the contribution is compared to existing literature.
Not for Theorems A, B, C, Lemma A there are missing citations. Are this results from reference [9]? Need to be improved.
There is no definition of the property (A) in introduction and in the manuscript.
There is no need for Section 2.
Line 33, there is written just “We”
In the manuscript is very short Discussion section, but there is a need to to discuss the results stated in the manuscript from the point of view of assumptions and theoretical limitations. Also the author should carefully state some general indications and next steps in the conclusion.
The paper cannot be published in its current form and I recommend MAJOR REVISION

Reviewer 4 Report
Comments and Suggestions for Authors
The paper focuses on the investigation of trinomial differential equations of the order n with deviating arguments. The properties of interest include the Property (A) (also referred to as Property A; in essence in means that if any of proper solutions is oscillatory when n is even and either is oscillatory or has a monotonically decreasing absolute value when n is odd; two independent sets of sufficient conditions are presented in Theorem 1 and Theorem 2) and its refinement which admits only oscillatory solutions (sufficient conditions are presented in Theorem 3).
The problem of finding oscillatory solutions is of essential interest in many applications (e.g., we tried to find such solutions to model the process of virus "hiding" and "returning", and argument deviation was one of the key model features). The results obtained by the author are new and non-trivial. The proofs are complete and correct.
There are several issues that may be addressed:
- from my point of view, it makes sense to illustrate formal definition and assertions with informal comments (e.g., to illustrate what Property (A) means, what is the essence and possible applications of Theorems);
- in my understanding, a criterion as a necessary and sufficient condition (actually, the results in the paper [9] by Koplatadze et al cited in the introduction as criteria in [9] are referred to as sufficient conditions); I highly recommend not to identify sufficient conditions and criteria, i.e., to replace all occurrences of the term "criteria" with "sufficient conditions";
- please remove the word "We" in line 33;
- Theorem B should end with a period, not with a ",." (line 39);
- the section "Materials and methods" is completely meaningless and can be removed;
- "a auxiliary" should be replaced with "an auxiliary" in line 50;
- since the formula (6) ends with a period, the first letter of line 52 should be capitalized;
- "An easy computations" in line 94 should be replaced either with "An easy computation" or with "Easy computations";
- it makes sense to add a comma after the word "respectively" in line 98;
- "y(t) is decreasing function" should be replaced with "y(t) is a decreasing function" between lines 117 and 118.
I think that after fixing these minor issues the paper can be published.
Comments on the Quality of English LanguageEnglish is quite understandable. A couple of minor issues are listed in Comments and Suggestions for Authors.
Round 2
Reviewer 2 Report
Comments and Suggestions for Authors
Accept after minor revision (corrections to text editing)
Comments on the Quality of English LanguageModerate editing of English language required
Reviewer 3 Report
Comments and Suggestions for Authors
Report
April 3rd, 2024
Paper: Property (A) and oscillation of higher-order trinomial differential equations with retarded and advanced arguments
Blanka Baculikova
Mathematics-2840468 (paper number)
Mathematics
Summary of the paper from the reviewer
The topic of the paper is interesting. The paper is improved. But there are some issues that should be changed.
The introduction should be reformulated, because of big rate of duplication.
Regardless the fact that you mentioned before Theorem A that criteria (not criterias) is from the reference [9], you can accidentally or intentionally cause confusion, so I strongly recommend that next to every cited Theorem should be added number of the reference.
The paper cannot be published in its current form and I recommend MINOR REVISION

Comments on the Quality of English LanguageAuthor Response
Please see the attachment.
